# BMP9 stimulates joint regeneration at digit amputation wounds in mice

Ling Yu[1], Lindsay A. Dawson[1], Mingquan Yan[1], Katherine Zimmel[1], Yu-Lieh Lin[1], Connor P. Dolan[1], Manjong Han[2] & Ken Muneoka[1,2]

A major goal of regenerative medicine is to stimulate tissue regeneration after traumatic injury. We previously discovered that treating digit amputation wounds with BMP2 in neo-natal mice stimulates endochondral ossification to regenerate the stump bone. Here we show that treating the amputation wound with BMP9 stimulates regeneration of a synovial joint that forms an articulation with the stump bone. Regenerated structures include a skeletal element lined with articular cartilage and a synovial cavity, and we demonstrate that this response requires the *Prg4* gene. Combining BMP2 and BMP9 treatments in sequence sti-mulates the regeneration of bone and joint. These studies provide evidence that treatment of growth factors can be used to engineer a regeneration response from a non-regenerating amputation wound.

[1] Department of Veterinary Physiology and Pharmacology, Texas A&M University, College Station, TX 77843, USA. [2] Department of Cell & Molecular Biology, Tulane University, New Orleans, LA 70118, USA. These authors contributed equally: Ling Yu, Lindsay A. Dawson. Correspondence and requests for materials should be addressed to K.M. (email: kmuneoka@cvm.tamu.edu)

Mammals display poor regenerative capabilities and the typical response to traumatic injury is fibrotic healing[1,2]. One strategy to enhance mammalian regenerative responses is to use developmental agents to re-initiate morphogenesis at non-regenerative amputation wounds[3], and this strategy has been used successfully to stimulate bone regeneration with targeted treatment of bone morphogenetic protein 2 (BMP2) or BMP7[4–8]. Transient treatment of the amputation wound with BMP2 induces formation of an endochondral ossification center that organizes the regeneration response of the amputated stump bone[4,5]. This induced regeneration response elongates the stump bone but fails to regenerate amputated distal structures such as the joint and the next distal skeletal element. A major next step in enhancing mammalian regeneration is to address regeneration of a joint, a structure that contains articular cartilage which displays poor regenerative capabilities[9]. During joint development the formation of the synovial cavity is central to joint morphogenesis[10], and synovial cavity formation fails in digits that lack the BMP2 antagonist Noggin[11], suggesting that regulation of BMP signaling is required for joint morphogenesis. To investigate joint morphogenesis and whether mammalian joints can regenerate, we have focused on a BMP family member, BMP9, which is specifically expressed by interzone cells of the forming synovial cavity and lacks a Noggin binding domain[12]. We hypothesize that BMP9 can act as a regenerative agent to stimulate joint regeneration at a non-regenerative amputation by re-initiating joint morphogenesis.

Here we demonstrate that treating non-regenerative digit amputation wounds with BMP9 stimulates the regeneration of joint structures. The regenerated joint includes a synovial cavity and a skeletal element lined with articular cartilage that articulates with the amputated bone stump. Further, we show that BMP9-induced joint regeneration requires the *proteoglycan 4* (*Prg4*) gene. Lastly, we demonstrate that the sequential treatment of BMP2 followed by BMP9 stimulates the regeneration of bone and joint by cells that would otherwise undergo fibrotic healing. Taken together, our findings demonstrate that growth factor treatment can be used to overcome mammalian regenerative defects, and provides further evidence that regenerative failure in mammals is associated with a defective wound environment.

## Results

**Exogenous BMP9 stimulates regeneration of joint structures**. *Bmp9* is predominately expressed by the embryonic liver, but during limb development, *Bmp9* is expressed by interzone cells during late stages of digit joint formation (Fig. 1a; Supplementary Fig. 1a). This expression is restricted to cells involved with cavitation to form the synovial cavity and is transient: *Bmp9* is not expressed prior to joint formation nor is it expressed after the joint is complete (Supplementary Fig. 1b-d). Since joint formation fails in mice lacking the BMP antagonist Noggin, and BMP9 is refractory to Noggin inhibition, its expression during joint morphogenesis raised the possibility that BMP9 can act as a regenerative agent to stimulate joint regeneration.

Digit amputation (Fig. 1b) is used as a non-regenerative model system in neonatal and adult mice[2,4,5,13]. Amputation bisects the sub-terminal phalangeal element (P2), removing the distal part of P2, the terminal phalangeal element (P3) and the intervening P2/P3 joint. Digit amputation results in skeletal truncation and fibrotic healing (Supplementary Fig. 1e) with no indication of a skeletal or joint regenerative response. At the core of induced regeneration studies is the view that cells of the amputation wound possess the necessary information to regenerate all structures removed by amputation, and we have shown that cells of the mouse amputation display this type of positional

information[4]. To treat amputation wounds with a purified growth factor, a single agarose bead is soaked in a high concentration of growth factor and implanted between the wound epidermis and the amputated stump bone. In vitro studies estimate that a single agarose bead delivers 5.5 ng of BMP9 that is released over a 72 h period (Supplementary Fig. 1f), and this delivery vehicle is intended to transiently stimulate cells locally at the amputation wound. BMP9 treatment stimulates the regeneration of a new distal skeletal element that appears to articulate with the stump bone (Fig. 1c) in 51% of the digits (49/96) analyzed using microcomputed tomography (μCT). This observation is significant as a distal skeletal element is not observed in any bovine serum albumin (BSA)-treated control digits (0/29 digits; $P$ < 0.0001). Digits evaluated by histological analysis, not μCT, demonstrate that the regenerated skeletal element is not ectopic, but forms a joint-like articulation with the stump bone that includes a distinct cavity associated with a chondrogenic layer of cells in 61% of the digits (60/98; Supplementary Fig. 2a, d, g, j). In digits maintained for an extended time period, the chondrogenic cell layer differentiates into articular cartilage that is histologically similar to undamaged articular cartilage (Fig. 1d, e). The BMP9-treated digit stump does not form a complete cartilage layer; however, patches of articular chondrocytes can be identified on the stump surface (Fig. 1e). The remaining BMP9-treated digits that lack a regenerated skeletal element (Supplementary Fig. 2b) appear similar to control BSA-treated digits which heal the stump bone with fibrotic tissue (Supplementary Fig. 2c).

**BMP9 induces cavitation and chondrogenesis**. Histological analysis 72 h following BMP9 treatment identifies two distinct responses: (1) digits that regenerated a chondrogenic nodule and cavity, referred to as cavity-forming digits (Supplementary Fig. 2d), or 2) digits that failed to form a nodule and cavity, referred to as non-cavity-forming digits (Supplementary Fig. 2e). The digit stump of BMP9-treated non-cavity-forming digits at 72 h appears undifferentiated and is capped with a cluster of chondrogenic cells (Supplementary Fig. 2e) that is maintained at 7 days (Supplementary Fig. 2h), but the digit stump is ossified by 14 days (Supplementary Fig. 2k). BMP9-treated cavity-forming digits maintain a chondrogenic nodule at 7 and 14 days following treatment (Supplementary Fig. 2g, j). The fibrotic healing response of amputated digits treated with BSA was similar to previous descriptions (Supplementary Fig. 2f, i, 1[1]).

The chondrogenic nature of the regenerated nodule in cavity-forming digits was confirmed based on expression of *Col2a1* transcripts 72 h after BMP9 treatment (Fig. 1f). We also observe expression of other chondrogenic genes known to be associated with the development of articular chondrocytes including *Sox9*[14], *Aggrecan* (*Acan*)[15], *Fibromodulin* (*Fmod*)[16] and *Upper growth plate and cartilage matrix associated protein* (*Ucma*)[17] (Fig. 1g–j). In addition, we show that doublecortin (*Dcx*)[18] is expressed by developing articular chondrocytes of the uninjured digit joint (Fig. 1k) and by surface cells of the BMP9-induced nodule 72 h after treatment (Fig. 1l), but not by cells of the BSA-treated digits (Supplementary Fig. 3a). Chondrogenic genes are not expressed by cells of the amputation wound in BSA-treated control digits (Fig. 1m–q). In BMP9-treated non-cavity-forming digits, we find transcripts of *Col2a1*, *Sox9, Acan, Fmod* and *Ucma* up-regulated at the distal end of the stump (Fig. 1r–v) at 72 h. This is consistent with histological studies (Supplementary Fig. 2e) and since BMP9-treated non-cavity-forming digits fail to maintain a chondrogenic layer of cells at the end of the stump (Supplementary Fig. 2b), the induced expression of these chondrogenic genes by BMP9 is not sufficient to drive articular cartilage regeneration on the digit stump. These studies show that BMP9 induces

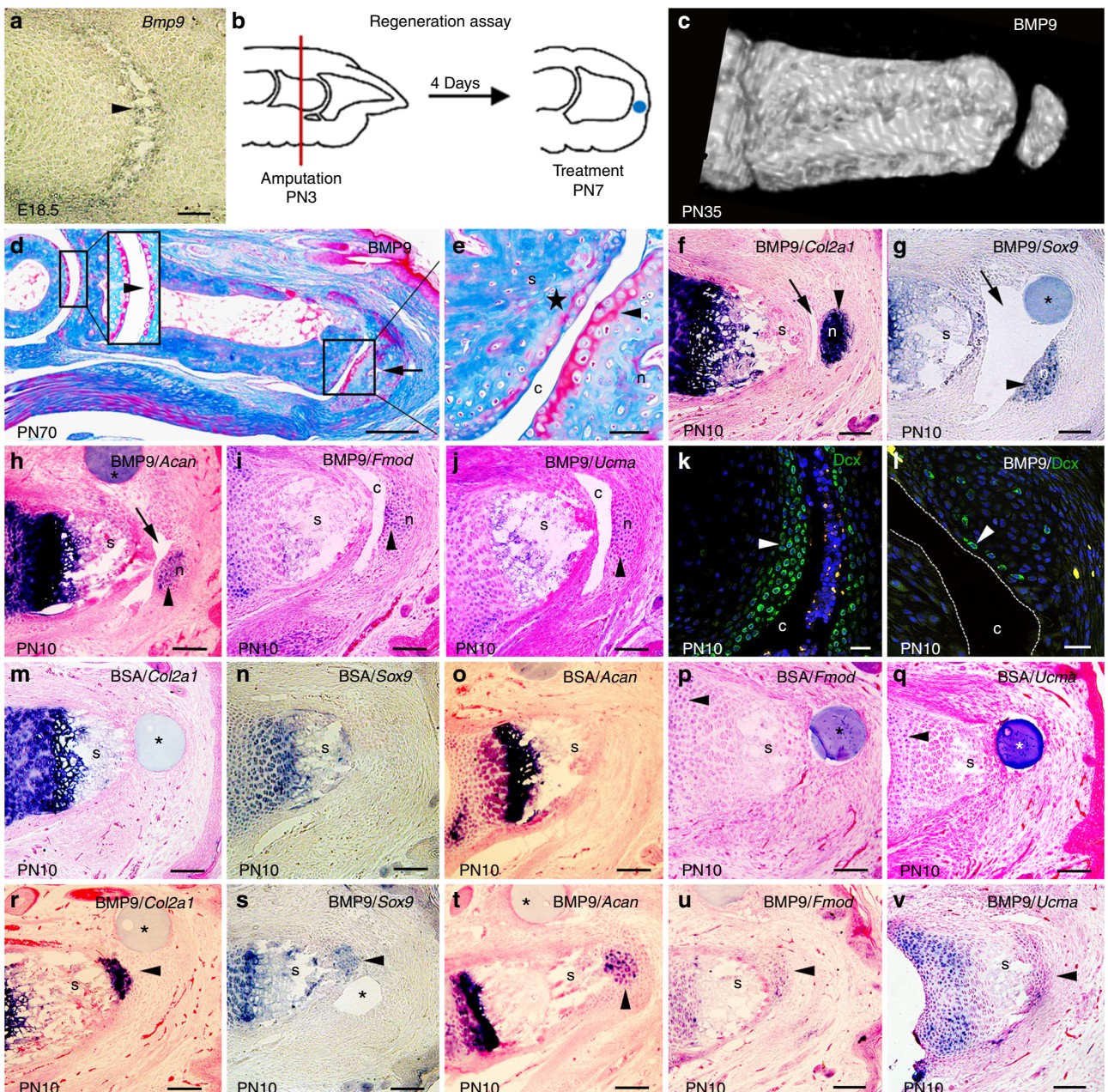

**Fig. 1** Bone morphogenetic protein 9 (BMP9) stimulates regeneration of joint structures. **a** *Bmp9* transcripts are localized to digit joints undergoing cavitation (arrowhead) in E18.5 embryos. **b** P2 level amputation of neonatal digits is used to test for induced regeneration. Following amputation (red line) at postnatal day 3 (PN3), epidermal closure is completed 4 days later (PN7), and an agarose microcarrier bead (blue dot) is implanted between the wound epidermis and the stump. **c** Microcomputed tomography (μCT) rendering of a BMP9-treated digit showing regeneration of a skeletal element articulating with the stump after 4 weeks (PN35). **d** Mallory trichrome staining of BMP9-treated digit 9 weeks after BMP9 treatment (PN70). Regenerated skeletal element (arrow) forms a joint-like structure with the digit stump. Articular cartilage of the P1/P2 joint is shown at higher magnification in the inset. **e** Higher magnification of the inset in (**d**) shows a cavity (c) separating a layer of articular cartilage (arrowhead) lining the regenerated skeletal element (n) and the stump bone (s). Small patches of articular cartilage (star) form on the stump bone. **f–j** In situ hybridization of cavity (arrow)-forming BMP9-treated digits showing localization of transcripts to the regenerated chondrogenic nodule but not the digit stump (s) 72 h after bead (*) implantation. **f** *Col2a1*. **g** *Sox9*. **h** *Acan*. **i** *Fmod*. **j** *Ucma*. **k, l** Immunofluorescence staining of Doublecortin (Dcx) expression. **k** Immunostaining shows Dcx+ cells (arrowhead) lining the synovial cavity of an uninjured joint at PN10. **l** Isolated superficial cells of the BMP9-induced nodule express Dcx (arrowhead) 72 h after treatment (PN10). The cavity is outlined with dashed white lines. Dcx+ cells are also observed internal to the superficial layer. **m–q** In situ hybridization of bovine serum albumin (BSA)-treated control digits 72 h after bead implantation showing no expression associated with the amputation wound distal to the stump. **m** *Col2a1*. **n** *Sox9*. **o** *Acan*. **p** *Fmod*. **q** *Ucma*. **r–v** In situ hybridization of non-cavity-forming BMP9-treated digits showing localization of transcripts to the distal stump 72 h after bead implantation. **r** *Col2a1*. **s** *Sox9*. **t** *Acan*. **u** *Fmod*. **v** *Ucma*. Right is distal, top is dorsal. Scale bars: **a, f–j, m–v** = 100 μm; **d** = 250 μm; **e, k** = 50 μm; **l** = 25 μm

chondrogenic gene expression in all treated digits, but only those digits that formed a cavity regenerate a joint-like structure that includes articular cartilage.

In BMP9-treated cavity-forming digits, Col2a1 expression is maintained by cells of the nodule 7 days after BMP9 treatment (Supplementary Fig. 3b), but these cells do not express Col10a1 transcripts (Supplementary Fig. 3c). Control BSA-treated digits at 72 h and 7 days do not express Col2a1 or Col10a1 (Fig. 1m, Supplementary Fig. 3d-f). ColX expression is observed at 11 days after BMP9 treatment (Supplementary Fig. 3g) and hypertrophic chondrocytes are identified histologically by 14 days (Supplementary Fig. 2j), indicating that the regenerated skeletal element forms by endochondral ossification. These data indicate that BMP9 stimulates the aggregation and differentiation of chondrocytes but their progression to hypertrophic chondrocytes is delayed compared to BMP2-treated digits[4]. Bromodeoxyuridine (BrdU) incorporation studies indicate that BMP9 stimulates proliferation of cells surrounding the bead at 72 h, as compared to BSA-treated controls, and that the proliferation response is independent of cavity formation (Supplementary Fig. 3h-j). In cavity-forming digits, the nodule is not next to the BMP9 bead and BrdU-labeled chondrocytes are not present, indicating that BMP9 does not stimulate chondrocyte proliferation (Supplementary Fig. 3h). The absence of proliferating chondrocytes and delayed hypertrophic differentiation distinguishes the BMP9 response from BMP2 stimulation of endochondral ossification[4].

***Prg4 is required for BMP9-induced joint cavation***. All BMP9-treated digits initiate a chondrogenic response, but only digits that also regenerate a cavity form a joint-like structure. This suggests that the cavitation response plays a key role in the joint regeneration response. Prg4 expression is associated with synovial cavity formation during development. Prg4 encodes for the core protein of lubricin, a proteoglycan found in synovial fluid, and its expression correlates with the early evolution of the vertebrate synovial joint[19]. Prg4 is expressed by the superficial cell layer of articular cartilage and recent studies show that stem/progenitor cells in this layer give rise to all articular cartilage layers[20,21]. Thus, the cavitation requirement in joint regeneration may be linked to progenitor cells associated with Prg4-expressing superficial cells. During digit development, Prg4 transcripts localize to the digit joint during cavitation (Supplementary Fig. 3k) and is specifically expressed by cells of the superficial cell layer lining the P2/P3 synovial cavity (Fig. 2a). Prg4 is not expressed by cells of BSA-treated control digit amputation (Supplementary Fig. 3l), or by BMP9-treated samples that fail to form a cavity at 72 h (Supplementary Fig. 3m). However, in BMP9-treated cavity-forming digits, Prg4 transcripts are expressed by cells of the induced cavity as well as by some cells of the distal nodule that abut the cavity (Fig. 2b). By 7 days after BMP9 treatment, only cells forming the cavity express Prg4, and the cavity abuts both the P2 stump and the regenerating nodule (Fig. 2c). The expression of Prg4 by cells forming the cavity identify the regenerated cavity as synovial[19]. At 24 h following BMP9 treatment, a clear cavitation response is not identified; however, we find samples in which clusters of Prg4-expressing cells are associated with regions of reduced cell density (Fig. 2d), suggesting a relationship between induced Prg4 expression and the initiation of cavitation.

To investigate if Prg4 is required for BMP9-induced cavitation, we performed P2 amputations and BMP9 treatment in the Prg4$^{-/-}$ mouse[22]. Prg4$^{-/-}$ mice form joints indicating that it is not required for joint development; however, mutant joints have reduced articular cartilage surfaces and display a joint degeneration phenotype[22,23]. Digits of Prg4$^{-/-}$ and Prg4$^{+/-}$ neonates

were amputated and treated with BMP9 or BSA after wound closure. BSA treatment did not evoke a regeneration response in Prg4$^{+/-}$ (n = 8 digits, Fig. 2e) or Prg4$^{-/-}$ (n = 4 digits, Fig. 2f) mice. As a positive control, Prg4$^{+/-}$ digits treated with BMP9 regenerated skeletal articulations at a frequency similar to that observed in wild-type mice (Fig. 2g, 50%; n = 24 digits). In contrast, Prg4$^{-/-}$ digits failed to display a BMP9-induced articulation response (Fig. 2h, 0%, n = 32 digits), demonstrating that the Prg4 gene is required for BMP9-induced regeneration of joint structures. The Prg4$^{-/-}$ response is reminiscent of BMP9-treated non-cavity-forming wild-type digits, and hence we investigated whether the BMP9-induced chondrogenic response was dependent on Prg4. Digits were amputated, treated with BMP9 after wound closure, collected 72 h later and analyzed by in situ hybridization. Prg4$^{-/-}$ digits treated with BMP9 failed to form a cavity, but cells at the distal stump did express transcripts for Col2a1, Sox9, Acan, Fmod and Ucma, indicating that the chondrogenic arm of the BMP9 response was intact and is not dependent on Prg4 (Fig. 2i–m). These studies demonstrate that BMP9 treatment of the digit amputation wound induces two distinct events, chondrogenic induction and cavitation. The induction of a chondrogenic response is consistently observed while BMP9-induced cavitation is more variable, and the data support the conclusion that both responses are required for the regeneration of joint structures.

**BMP9 stimulates joint regeneration in adult digits**. Developing tissues are known to display enhanced regenerative/repair properties when compared to mature adult tissue. To explore the effect of BMP9 treatment on adult digit amputations we used a sol–gel delivery mechanism effective in eliciting a BMP2-induced bone regenerative response[5]. Using this protocol, the effect of BMP9 on adult digit amputations was tested by applying BMP9 (250 ng) 9 days after amputation. BSA-treated control digits healed without a regenerative response as previously described[5]. Digits were collected 7 and 14 days after BMP9 treatment and were analyzed for a regenerative response. BMP9 induced a robust chondrogenic response in all of the digits analyzed at 7 days (n = 4). The regenerated cartilage grew well beyond the amputation plane (Fig. 3a) forming a mushroom-shaped cartilage mass that was contiguous with the peripheral callus that forms after digit amputation[1]. The large cartilaginous callus is composed of chondrocytes embedded in a dense extracellular matrix (Fig. 3b) and cells are immunopositive for Sox9 and Acan (Fig. 3c, d). We did not observe a cavity associated with the chondrogenic response in our samplings at 7 days. The majority of digit samples analyzed at 14 days (n = 4/5) showed a similar but enlarged chondrogenic response compared to digits collected at 7 days. However, one of the digits displayed a milder chondrogenic response that was not associated with the peripheral callus but was distal to the amputated stump and formed a cavity between the stump and the regenerated chondrogenic element (Fig. 3e, f). The stump component of the cavity was differentiated bone, while the cells lining the regenerated distal element were positive for Sox9 and internal cells expressed both Sox9 and Acan (Fig. 3g, h). The regenerated structure was immunonegative for the developmental articular chondrocyte marker, Dcx (Supplementary Fig. 3n). These studies indicate that BMP9 stimulates both a chondrogenic and cavitation response at adult digit amputation wounds, but the chondrogenic arm of the BMP9 response dominates. Nevertheless, the data show that joint-like structures can be induced to regenerate in adult mice.

**Combined regeneration of bone and joint with BMP2 and BMP9**. The digit synovial joint is a tri-layered structure consisting

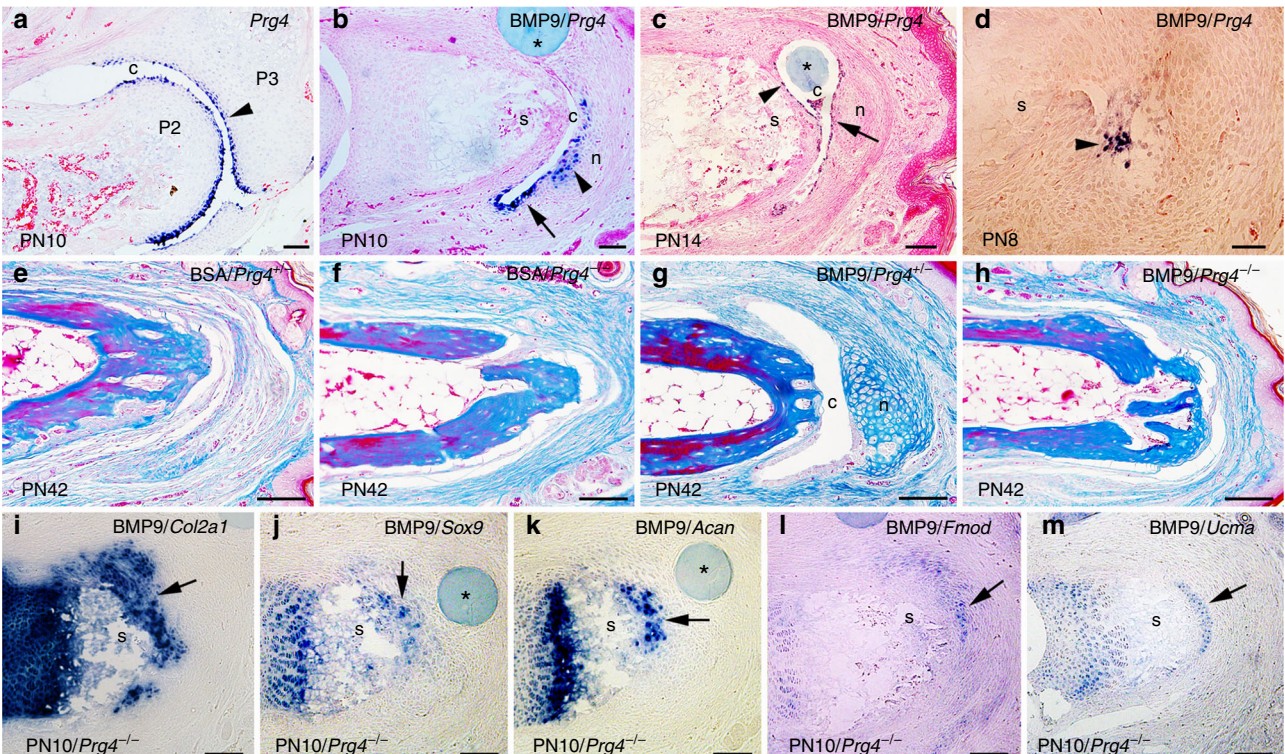

**Fig. 2** *Proteoglycan 4 (Prg4)* is required for bone morphogenetic protein 9 (BMP9)-induced cavitation. **a–d** In situ hybridization for *Prg4* expression. **a** *Prg4* is expressed by cells lining the P2/P3 synovial cavity (arrowhead) and forming the superficial layer of the articular cartilage of the neonatal joint at PN10. **b** In BMP9-treated cavity-forming digits, *Prg4* is expressed by cells lining the induced cavity (arrow) and by cells forming the chondrogenic nodule (arrowhead) 72 h after treatment. **c** At 7 days after treatment with a BMP9 bead, *Prg4* is expressed by cells lining the cavity on both the stump side (arrowhead) and the nodule side (arrow). **d** At 24 h after BMP9 treatment, *Prg4* transcripts are found localized in clusters of cells in the wound bed (arrowhead) associated with the initiation of the cavitation response. **e, f** Control bovine serum albumin (BSA) bead implantation does not induce an articulation response in *PRG4+/−* (**e**) or *PRG4−/−* (**f**) digit amputations. **g** BMP9 treatment of *PRG4+/-* digit amputations induces an articulation response that includes a distal nodule and cavity. **h** No regeneration response is observed after BMP9 treatment in *PRG4−/−* mutants. **i–m** In situ hybridization of chondrogenic genes expressed at the distal stump in non-cavity-forming *PRG4−/−* mutant digits 72 h after BMP9 treatment. **i** *Col2a1*. **j** *Sox9*. **k** *Acan*. **l** *Fmod*. **m** *Ucma*. Distal is to the right and dorsal is to the top; n nodule, c cavity, s stump, * indicates bead. Scale bars: **a–d**, **i–m** = 100 μm; **e–h** = 200 μm

of a central cavity with two articular cartilage layers associated with the adjoining bone elements. BMP9 treatment of neonatal digit amputations result in the formation of a synovial cavity and a distal, but not proximal, articular cartilage layer. At the time of amputation, the P2 stump is undergoing ossification and we hypothesize that the differentiation state of the stump limits the response to BMP9. To explore this we took advantage of our previous finding that BMP2 stimulates the formation of an apical population of proliferating chondrocytes that establishes an endochondral ossification center[4]. In the BMP2 response the apical zone of proliferating chondrocytes are present from 3 to 7 days after BMP2 treatment. Therefore, we carried out studies in which the amputation wound is treated sequentially; first, with BMP2 to induce proliferating chondrocytes and, second, with BMP9 to stimulate joint regeneration (Fig. 4a). These sequential treatment studies (B2–B9) were carried out with a 3- or 7-day interval separating the two treatments to target the proliferating chondrocyte cell population. The 3-day interval stimulated joint regeneration at a frequency similar to BMP9 alone (57/95 (58%), Fig. 4b) and the 7-day interval stimulated regeneration at a significantly higher frequency (56/80 (70%), *P* = 0.0135). In both series, histological analyses indicated the regeneration of a complete joint structure that included a layer of articular cartilage lining the stump (Fig. 4c). Control studies in which BMP2 treatment was followed by a second treatment with another BMP2 bead stimulated excessive bone growth but not the regeneration of joint structures (Fig. 4d; 0/16 digits). In another

control series, amputated digits were treated with BMP9 10 days after amputation and these digits regenerated joint structures that did not involve the digit stump (Fig. 4e, f). Comparing the sequential B2–B9 response with control undamaged digit joints shows that the regenerated cavity consisted of opposing layers of cells that are immunopositive for ColII (Fig. 4g, h), Acan (Fig. 4i, j) and Dcx (Fig. 4k; Fig. 1k), consistent with the regeneration of articular cartilage. The expression of *Prg4* transcripts 96 h after B2–B9 treatment confirmed the regeneration of a synovial cavity (Fig. 4l), and expression of *Fmod* and *Ucma* by cells of the distal stump confirmed the differentiation of chondrocytes (Fig. 4m, n).

B2–B9-treated digits displayed an enhanced joint regeneration response (i.e., two articular cartilage layers) and a higher response frequency, suggesting that BMP2 treatment elevated the BMP9 cavitation response. To explore this further, 72 h after treatment with BMP2, we found a robust *Prg4* expression domain associated with the BMP2 bead but without a cavitation response (Supplementary Fig. 3o), indicating that both BMP2 and BMP9 induce *Prg4* expression by cells of the amputation wound. However, cavitation only occurs following BMP9 treatment, indicating that *Prg4* expression is required but not sufficient to induce cavitation and joint regeneration, and suggests that BMP2 enhances the BMP9 response by enrichment of the *Prg4*-expressing cell population. Since BMP9 treatment alone stimulated chondrogenesis but delayed hypertrophic differentiation, we explored whether BMP2-induced endochondral ossification of the stump occurs in conjunction with joint regeneration following

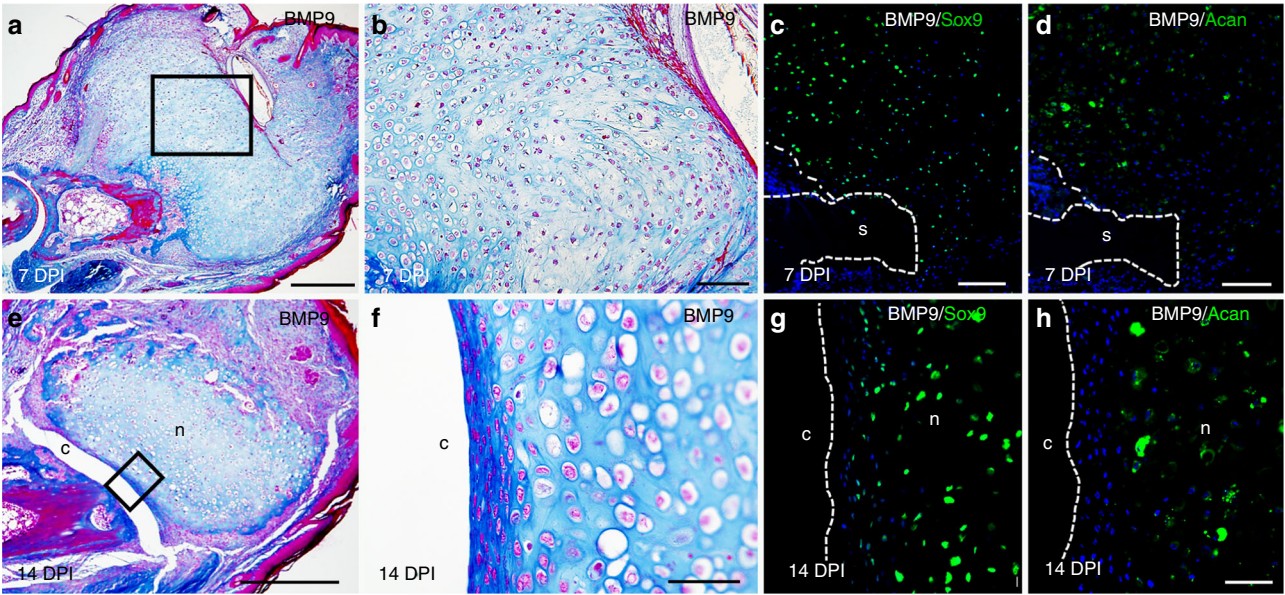

**Fig. 3** Bone morphogenetic protein 9 (BMP9) stimulates stump chondrogenesis and cavitation in adult digits. **a** Mallory trichrome staining of an adult BMP9-treated digit harvested at 7 days post implant (DPI) showing induced chondrogenesis is contiguous with the bone stump. **b** Higher magnification of the inset in (**a**) of BMP9-induced chondrocytes at 7 DPI. **c**, **d** Immunofluorescence staining for the cartilage marker proteins Sox9 and Acan illustrate chondrogenesis contiguous with the stump bone (dashed white lines) at 7 DPI (n nodule, c cavity, s stump). **e** Mallory trichrome staining of an adult BMP9-treated digit showing cavity and chondrogenic nodule formation at 14 DPI. **f** Higher magnification of the inset in (**e**) showing the chondrogenic and cavitation response at 14 DPI. **g**, **h** Immunohistochemistry of the 14 DPI cavity-forming BMP9-treated digit. Dashed white lines outline the cavity. **g** Cells lining the cavity and within the nodule express the cartilage marker protein Sox9. **h** Acan immunostaining is restricted to cells within the central region of the nodule. Distal is to the right and dorsal is to the top. Scale bars: **a**, **e** = 500 µm; **b**, **c**, **d** = 100 µm; **f**–**h** = 50 µm

B2–B9 treatment. *Col2a1*, but not *Col10a1*, transcripts are localized to distal stump cells 48 h after BMP9 treatment consistent with an inhibitory effect of BMP9 on hypertrophic chondrocyte differentiation (Fig. 4o, p). At 96 h, however, both *Col2a1* and *Col10a1* transcripts co-localize to cells of the digit stump (Fig. 4q, r), indicating that endochondral ossification of the stump does occur in conjunction with BMP9-stimulated joint regeneration. Endochondral ossification of the stump was also confirmed based on histological analysis (Supplementary Fig. 3p), and direct measurements of the P2 stump indicated significant skeletal elongation of B2–B9-treated digits compared to BSA controls (B2–B9 = 1.28 mm, $n = 32$, s.d. = 0.15 vs. BSA = 0.86 mm, $n = 21$, s.d. = 0.14; $P < 0.01$). These results demonstrate that BMP2 treatment of the amputation wound stimulates formation of an endochondral ossification center that mediates skeletal growth from the stump, and also establishes a responsive population of apical cells that can participate in joint regeneration if appropriately stimulated. Thus, sequential treatment of the amputation wound with BMP2 and BMP9 successfully integrates bone regeneration with joint regeneration.

## Discussion

BMP9 is a member of the BMP family of growth factors[24] produced by the liver[25] and present in circulating plasma[26]. BMP9 is a physiological ligand for activin receptor-like kinase 1[27] and, like other BMP family members, signals through the phosphorylation of Smad 1/5/8[28]. Unlike other BMP family members, BMP9 is not effected by BMP antagonists such as Noggin[12] which suppresses BMP signaling and is necessary for joint morphogenesis[11]. We show here that *Bmp9* is transiently expressed during digit joint cavitation and that BMP9 treatment stimulates joint regeneration. Mutant studies indicate that *Bmp9* is not essential for joint formation[29]; however, redundancy of gene function is frequently observed in joint and skeletal development[30–33] and regenerative

events are stimulated by treatment with non-essential gene products[4,8,34]. BMP9 has been shown to play a role in the regulation of angiogenesis[35], liver fibrosis[36], osteogenesis[37] and chondrogenesis[38–41]. During angiogenesis, BMP9 acts in a context-dependent manner, inhibiting neovascularization in a model of age-related macular degeneration[42], while stimulating angiogenesis associated with induced osteogenesis[43]. Similarly, BMP9 inhibits endogenous digit tip regeneration in neonatal mice[44], while stimulating the regeneration of joint structures at non-regenerative amputations. Such context-dependent responses indicate that cells associated with the injury response determine the outcome of BMP9 stimulation, thus shifting attention to the cell types participating in endogenous repair/regenerative programs.

The joint is an essential structural component of the musculoskeletal system, but unlike bone and skeletal muscle, joint tissues, particularly articular cartilage, display a limited capacity for repair[9]. Joint degenerative disorders such as osteoarthritis affect a large portion of the world population and most clinical approaches focus on engineering articular cartilage for transplantation, not on enhancing endogenous regenerative properties[45]. We have discovered that treating amputation wounds with BMP9 stimulates the regeneration of joint structures, including articular cartilage and synovial cavity. The action of BMP9 on joint regeneration is likely indirect since two independent induction events, chondrogenesis and cavitation, are required. Further, BMP9 morphogenetic activity can occur at a distance from the BMP9 source, while proliferation is stimulated in neighboring cells that are not involved in either response. The requirement of *Prg4* by cells undergoing cavitation is consistent with lineage studies identifying stem cells in the superficial layer associated with the synovial cavity as a cell source for all layers of articular cartilage[20,21]. The differential response of the digit stump versus the regenerated skeletal element suggests that BMP9-induced chondrogenesis plays a supporting role for

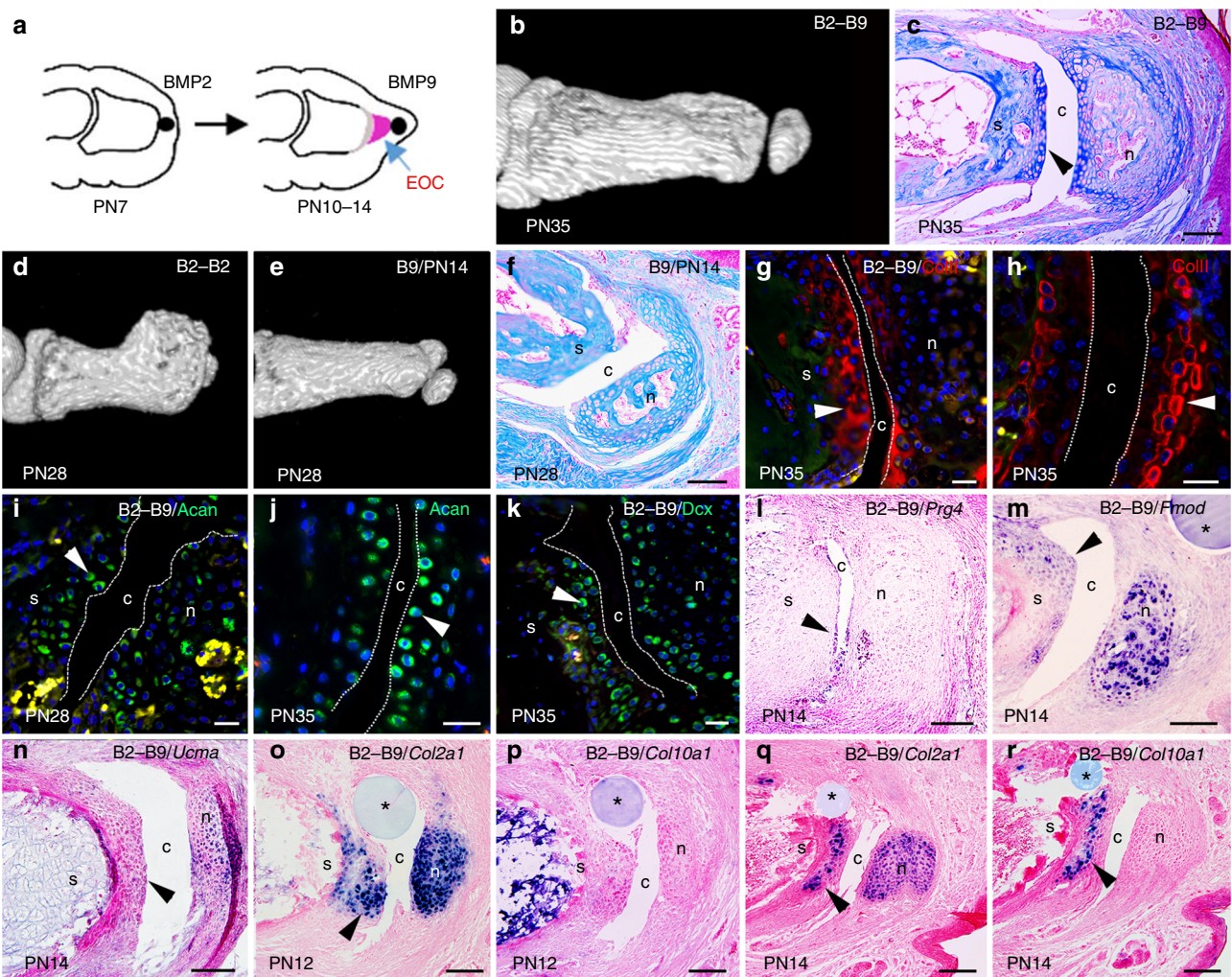

**Fig. 4** Sequential bone morphogenetic protein 2/9 (BMP2/BMP9) treatment stimulates bone and joint regeneration. **a** Diagram illustrating sequential BMP2 and BMP9 (B2–B9) treatment. A BMP2 bead is implanted at PN7. A BMP9 bead is implanted distal to the induced endochondral ossification center (EOC) at PN10 or PN14. **b–m** All B2–B9 samples are from the 3-day interval study. **b** microcomputed tomography (μCT) rendering of a B2–B9-regenerate at PN35 identifies a distal articulation. **c** Mallory trichrome staining of a B2–B9 regenerate showing cartilage capping the stump and surrounding the nodule (n nodule, c cavity, s stump). **d** μCT rendering of a PN28 regenerate treated with BMP2 at PN7 and PN10 displaying excessive bone growth and no joint formation. **e** PN28 μCT rendering of a regenerate treated with BMP9 at PN14. Thus, 50% of the digits formed a nodule that articulated with the stump. **f** Mallory trichrome staining of digit shown in (**e**). The regenerate formed a cavity and a cartilage-lined nodule but the stump bone lacks a chondrogenic cell layer. **g–k** Immunofluorescence staining for marker proteins expressed by cells lining both sides of regenerated and control joints. The joint cavity is identified by dashed white lines. Red blood cell autofluorescence (yellow) identifies ossifying regions. **g** ColII (arrowhead) of the B2–B9 induced cavity and **h** a control digit joint cavity. **i** Acan (arrowhead) of the B2–B9-induced cavity and **j** a control digit joint cavity. **k** Dcx (arrowhead) of the B2–B9-induced cavity. **l–r** In situ hybridization of B2–B9 regenerates (* indicates bead). **l** *Prg4* transcripts are expressed by cells of the regenerated cavity (arrowhead) 96 h after BMP9 treatment. **m**, **n** *Fmod* (arrowhead in **m**) and *Ucma* (arrowhead in **n**) transcripts are expressed in cells of the distal digit stump and the nodule 96 h following BMP9 treatment. **o**, **p** Cells of the distal stump (arrowhead) and the regenerated nodule express *Col2a1* (**o**), whereas *Col10a1* is not expressed (**p**) 48 h after BMP9 treatment. **q**, **r** Cells of the distal stump (arrowhead) and the regenerated nodule express *Col2a1* (**q**) but only cells of the stump express *Col10a1* (**r**) 96 h after BMP9 treatment. Right is distal, dorsal is top. Scale bars: **c** = 200 μm; **g-k** = 50 μm; **f**, **l-r** = 100 μm

articular cartilage differentiation by superficial chondrocytes. This conclusion is further supported by B2–B9 studies showing that chondrogenic modification of the stump by BMP2 enhances the BMP9 response. Thus, the endochondral ossification center established by BMP2 generates a chondrogenic template that supports osteoblast differentiation at one end[4,5], and articular cartilage differentiation at the other end, and both processes can be stimulated at a non-regenerative amputation wound.

Transient treatment with either BMP2 or BMP9 transforms the mammalian wound healing response in distinct ways to stimulate regeneration. In the case of BMP2, a zone of proliferating chondrocytes differentiate into hypertrophic chondrocytes that

forms a template for stump bone regeneration[4]. BMP2 also enhances cell recruitment by activating the SDF-1/CXCR4 signaling pathway;[46] thus, induced skeletal regeneration involves a coordinated response by multiple cell types in the amputation wound. In the case of BMP9 treatment, a coordinated response involving condensation and differentiation of chondrocytes coupled with the formation of a synovial cavity results in the regeneration of joint structures. These two regenerative responses identify discrete morphogenetic programs that can be activated either independently or coordinately within the context of the amputation wound. These findings provide further support for the view that cells of a non-regenerative mammalian

amputation wound retain the positional information necessary to re-build structures removed by amputation, and that regenerative failure results from a deficiency of the wound environment that can be extrinsically manipulated to engineer regenerative events.

## Materials and methods

**Mouse digit amputation and BMP treatment.** Pregnant outbred CD1 mice were purchased from Harlan Laboratories (Indianapolis, IN). The *Prg4* knockout mice (*Prg4tm1Mawa/J*) were purchased from the Jackson Laboratory (Bar Harbor, ME). For neonatal studies, digit amputations were carried out on postnatal day 3 (PN3), and treatment of the amputation wound was carried out after wound closure on PN7 as previously described[4]. Briefly, a single agarose microcarrier bead (Affi-Gel Blue Gel beads, Bio-Rad, Hercules, CA) was used to deliver BMP9, BMP2 or BSA to the amputation wound. Microcarrier beads were soaked with recombinant human BMP9 (500 ng/μl; R&D Systems, Minneapolis, MN), recombinant human BMP2 (500 ng/μl; R&D Systems, Minneapolis, MN) or BSA (0.1% in phosphate-buffered saline) and implanted between the bone stump and wound epidermis as previously described[8]. For sequential growth factor treatment, amputated digits were treated with a BMP2 bead at PN7 to induce chondrogenic condensation and then treated with a BMP9 bead 3 (PN10) or 7 (PN14) days later. A minimum sample size ($n = 8$) was determined via power analysis ($1 - \beta = 0.8$; $\alpha = 0.05$). The amount of BMP9 delivered by a single agarose bead was estimated in vitro using the BRITER BMP responsive cell line (Kerafast, Inc., Boston, MA). BRITER cells contain a luciferase reporter under the control of the *Id1* promoter and provides a readout of Smad mediated signaling. Since BMP9 lacks a Noggin binding site, BRITER cells were used in the presence of Noggin (100 ng/ml; R&D Systems, Minneapolis, MN) to measure BMP9 signaling without background BMP2/4/7 signaling. Different doses of BMP9 in media containing 0.1% fetal bovine serum were used to generate a standard response curve from 0.1 to 50 ng/ml. BRITER cells were incubated with BMP9 for 5 h and the Luciferase Assay System (Promega Corp., Madison, WI) was used to measure luciferase activity. To estimate agarose bead delivery, 30 BMP9 beads were sequentially placed in 1 ml of assay media for 6, 18 and 48 h and luciferase activity was measured. BMP9 released from the beads was determined by comparison to a BMP9 standard response curve. All measurements were carried out in triplicate.

For adult studies, digit amputations were carried out on 2–3-month-old mice as previously described[1]. The amputation wound was immediately closed with a ventral skin flap that included the P3 fat pad and sealed with Dermabond (Ethicon, Somerville, NJ). BMP treatment was performed using a sol–gel silica-based glass with embedded growth factor implanted between the bone stump and distal skin as previously described[5]. Amputated digits received a single treatment of BMP9 (250 ng) on 9 days post amputation (9 DPA) and digits were analyzed 7 (16 DPA) and 14 (23 DPA) days later. All studies comply with relevant ethical regulations for animal testing and research, and received ethical approval by the Institutional Animal Care and Use Committees at Tulane University and Texas A&M University.

**Tissue processing.** Digits were fixed in buffered zinc formalin (Z-Fix, Anatech Ltd, Battle Creek, MI; overnight at room temperature) for histology and immunohistochemistry, or 4% paraformaldehyde (4 ºC, overnight) for section in situ hybridization. Decalcification was carried out using Decalcifier I (Surgipath, Leica Biosystems, Richmond, IL). Tissues were processed for paraffin histology as previously described[47] and stained with Mallory Trichrome[48] or Picro-sirius red[49]. BrdU labeling to identify proliferating cells was performed as previously described[8]. For section in situ hybridization, antisense riboprobes were generated using the Digoxigenin-UTP transcription labeling Kit (Roche, Indianapolis, IN) as previously described[47]. The following complementary DNA fragments were used to generate antisense riboprobes: *Bmp9* (707 bp), *Prg4* (600 bp), *Ucma* (363 bp), *Fmod* (497 bp), *Type II Collagen* (*Col2a1*, 500 bp), *Type X Collagen* (*Col10a1*, 650 bp) and *Sox9* (586 bp). The 2–5 digit samples were analyzed for each time point. Control studies included BSA-treated amputated digits or stage-matched unamputated digits. Sections were either not counterstained or stained with Alizarin Red (Sigma-Aldrich Co., St Louis, MO) and mounted with Permount Mounting Medium (Thermo Fisher Scientific, Waltham, MA). Imaging of histological and in situ hybridization sections was performed using the Olympus BX60 microscope and DP72 camera, with the DP2-BSW software (Olympus America Inc., Center Valley, PA). For immunohistochemical staining, paraffin sections were deparaffinized and rinsed in Tris-buffered saline with Tween® 20 (Sigma-Aldrich Co.). Antigen retrieval was performed using heat (90 ºC, 20 min, pH 6 citrate buffer) or proteinase K (10 mg/ml, 27 ºC, 12 min, Dako, Carpinteria, CA). Sections were incubated in a blocking solution (1 h, room temperature, Dako) followed by primary antibody (overnight, 4 ºC) and secondary antibody (45 min, room temperature). Antibody details are as follows: Collagen II: mouse anti-ColII (Acris Antibodies, San Diego, CA; AF5710, 1:200 dilution), proteinase K, Alexa Fluor® 568 goat anti-mouse IgG secondary antibody (Invitrogen, Carlsbad, CA; AF11004, 1:500 dilution); Collagen X: rabbit anti-ColX (Abcam, Cambridge, UK; AB58632, 1:500 dilution), proteinase K, Alexa Fluor® 488 goat anti-rabbit IgG secondary antibody (Invitrogen, Carlsbad, CA; A11008, 1:500 dilution); Aggrecan: rabbit anti-Acan (EMD Millipore, Billerica, MA; AB1031, 1:300 dilution), heat retrieval, Alexa Fluor® 488 goat anti-mouse IgG

secondary antibody (Invitrogen, Carlsbad, CA; A11008, 1:500 dilution); Doublecortin: rabbit anti-Dcx (Abcam, Cambridge, UK; AB207175, 1:200 dilution), heat retrieval, Alexa Fluor® 488 goat anti-rabbit IgG secondary antibody (Invitrogen, Carlsbad, CA; A11008, 1:500 dilution); Sox9: rabbit anti-Sox9 (Abcam; AB185966, 1:500 dilution), heat retrieval, Alexa Fluor® 488 goat anti-mouse IgG secondary antibody (Invitrogen; A11008, 1:500 dilution). Immunostained sections were counterstained with 4′,6-diamidino-2-phenylindole (DAPI; Invitrogen). Slide imaging was performed with the Olympus BX61 fluorescence deconvolution microscope and Slidebook software (Intelligent Imaging Innovations, Inc., Denver, CO).

**Microcomputed tomography scans and image processing.** To screen for element regeneration after BMP treatment, P2 digits were scanned at PN28, PN35 or PN42 using the vivaCT 40 (SCANCO Medical, Wayne, PA) as previously described[50]. Digit samples were scanned using high-resolution settings (voxel size 10.5 μm and energy of 55 kVp; 1000 projections per 180º captured at 380 msec using continuous rotation). μCT files were saved as a dicom image stack, and subsequently uploaded to ImageJ to generate images. The BoneJ[51] (Version 1.2.1) Optimized Threshold Plugin for ImageJ was used to create three-dimensional renderings. The Mann–Whitney test was used to determine statistical significance for induced -cavity formation after BSA, BMP9, or BMP2+BMP9 treatment. Statistical analysis was performed using GraphPad PRISM (GraphPad Software, La Jolla, CA).

**Reporting summary.** Further information on experimental design is available in the Nature Research Reporting Summary linked to this article.

## Data availability
All relevant data are available from the authors upon request.

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

## Acknowledgements

We thank Regina Brunauer and Rosalie Anderson for comments on the manuscript, and members of the Muneoka Laboratory (past and present) for thoughtful discussions. We thank the referees for their thoughtful comments. The research is funded by W911NF-06-1-0161 from DARPA, W911NF-09-1-0305 from the US Army Research Center, the John L. and Mary Wright Ebaugh Endowment Fund at Tulane University and Texas A&M University.

## Author contributions

Experimental design: L.Y., M.H., L.A.D. and K.M.; animal studies: L.Y., M.Y. and L.A.D.; histology: K.Z., Y.-L.L. and C.P.D.; immunohistochemistry: M.Y., L.A.D. and C.P.D.; in situ hybridization: L.Y. and M.Y.; data analysis: L.Y., M.H., L.A.D. and K.M.; manuscript writing: K.M., L.Y. and L.A.D.

## Additional information

**Competing interests:** K.M., L.Y. and M.Y. disclose patent no. 9,833,481 B2 (4104-1) on BMP9-induced articular cartilage regeneration. The authors declare no competing interests.

