## [Peer Review File · Nature Communications]

Reviewers' Comments:

Reviewer #1:

Remarks to the Author:

General comments:

This MS contends that BMP9 treatment of digit amputation wounds in mice stimulates regeneration of synovial joint structures that articulate with the stump bone. As previous studies showed that treatment by other BMPs induced endochondral bone formation but did not result in regeneration of joint structures, this finding, if accurate, represents a novel and perhaps clinically significant finding. However, while interesting, the data provided are incomplete and many of the appropriate controls required to verify the specific actions of BMP9 are missing. In addition, the authors provide no mechanism by which BMP9, unlike other BMPs, would be able to initiate synovial joint regeneration.

Specific comments:

1. The ISH for Bmp9 are really difficult to see; Fig 1A shows many areas of potential Bmp9 expression throughout the section and nothing that appears specific to the interzone. As Bmp9 is not expressed in these other areas, it is possible that the reported interzone staining is an artifact. As Bmp9 KO mice have no skeletal phenotype, Bmp9 is certainly not required for digit joint formation. The authors are encouraged to use other validation methods (laser capture microdissection followed by qPCR for example) to verify the localization of Bmp9 in the digit joint.

2. The authors state that they see joint regeneration 51% of the time suggesting the repair observed is quite variable. The authors should show the best and worst case scenarios and provide the criteria they used to score each of the digit regenerates.

A complete time course comparing control to BMP9 treated digit stumps should also be included so that the full picture of BMP9 activity can be evaluated. As BMP9 is known as a potent bone formation agent, more potent than BMP2, the authors should comment on whether there was any ectopic bone formation that occurred.

3. For the sequential bead implantation studies, several additional controls should be included; BMP2 followed by BMP2; BMP9 followed by BMP9; and those should be compared to BMP2 followed by BMP9 and BMP9 followed by BMP2. These data would solidify the unique role of BMP9 and remove the possibility that continuous BMP activity, unrelated to BMP9, is what drives the regenerate.

4. The authors provide no mechanism for why BMP9 might have a unique activity but do suggest it might be due to utilization of Alk1 and Acvr2B. Since there are good reagents to localize these receptors, and there are cKO mice as well, the authors could explore their hypothesis by localizing the receptors during the regeneration process and demonstrating regeneration doesn't occur in their absence or when other BMPs are implanted.

Reviewer #2:

Remarks to the Author:

This paper continues with this group's previous studies on how to stimulate regeneration of digits (and lower limbs) in mice. They have previously shown that a P2 amputation of a digit does not regenerate anything but if BMP2 is added on a bead then growth of the phalange is stimulated and new endochondral bone is formed. There is no new joint, however.

This work shows that by the subsequent addition of BMP9 (after BMP2) a new joint can be produced in addition to the elongation of P2. Making a new joint is a major achievement and represents a highly significant advance by this group.

The joint is demonstrably so because of i) articular cartilage on both the end of P2 and the 'nodule'; ii) the expression of several joint markers (Fmod, Ucp1, Acan, Dcx); iii) the failure to form in a Prg4 mutant mouse.

I think this work should be published in Nature Comm.

I have a few comments on the data.

1. Have these BMP2/BMP9 experiments been done on adults which would be directly relevant to

adult human digit amputations? In this work the digits are amputated on PN3.

2. line 54/55. In Fig.1F there is articular cartilage on the nodule. Why is there no articular cartilage on the nodule in Fig. 1G ? Is it because of the word 'occasionally' in line 53?

3. Most of the in situs show very clear signals, but a few of them do not. I find it very difficult to see the signal in SFig 1N & O (Fmod & Ucma respectively).

4. lines 83/84. Fmod and Ucma seem to be up-regulated throughout the nodule rather than just in the cells on the joint side of the nodule. Is the whole nodule a joint or is this what happens in the development of a joint?

5. Fig. 1P - there looks to be 1 Dcx +ve cell on the P2 side of the joint, agreed it is not on the endochondral surface. Are there non joint cells that express Dcx?

6. As you read this you get the impression that BMP2 is going to lengthen the P2 phalange, then BMP9 is going to make a joint. So the final product should be quite a bit longer than either a control (nothing) or a BMP9 only (just a joint) regenerate. But then when I compare Fig. 1C with Fig. 3B they both look the same. Have I got the wrong impression or is there not an overall difference in these varying regenerates?

Reviewer #3:

Remarks to the Author:

Previous work by this group has demonstrated that digit tip regeneration can be induced in the early post-natal mouse with individual morphogenetic agents (e.g., BMP2 and BMP7), which initiate a "multi-tissue response that culminates in structural regeneration" (Stem Cells Transl. Med., 2018). The goal in this manuscript was to test whether BMPs could induce both bone and cartilage formation and in doing so, regenerate the joint space.

The basis of this work is, I believe, predicated on Urist's findings in the early 1970's that BMPs induce chondrogenesis. Here, the authors use a BMP9 soaked bead to also induce chondrogenesis, this time at the digit tip. I don't think its correct to state, "This (chondrogenic) response is specific to BMP9 treatment" since they (and others) have shown that other BMPs can also induce chondrogenesis (e.g., Yu et al., Dev Biol, 2012). What is new, however, is the observation that BMP2 and BMP9 have differing effects on the wound blastema: where BMP2 induces the formation of cartilage that undergoes hypertrophy, BMP9 induced cartilage does not.

What is missing from this otherwise compelling study is a clear mechanism of action to explain the observed phenomenon. Other investigators have shown that BMP9 induces hypertrophic-like state with increased collagen type X expression in chondrocytes (van Caam et al. 2015), and that BMP9 can directly induce osteogenesis (Fujioka-Kobayashi et al. 2018). Presumably the cellular response to BMP9 depends in part on the identit(ies) of the cells on which it is acting. Which cells are responding to BMP9 in the digit anlagen? Given that chondroblasts and osteoblasts arise from a common progenitor (e.g., see Nature Cell Biology (Ono et al. 2014), it is unclear how the authors' findings align with the current literature. The experiments conducted with the Prg4 mutant strain were not particularly informative with regards to a mechanism of action. There are other questions (outlined below) that if answered, would significantly raise enthusiasm for this interesting paper.

1. The authors should provide some justification as to why BMP9 was chosen to stimulate joint formation.

2. Given the clinical side effects of BMP use (Carragee 2014), it would be helpful to include information on dose-dependent effects in this developmental context.

3. Questions about dose: why did the authors choose a bead soaking concentration of 0.5ug/ul? What was the actual dose delivered to the tissues? What is the half-life of BMP2? The reason why this latter question is important to answer is based on the observation that sequential treatment with BMP2 followed by BMP9 appeared to require an interval of 3 or 7 days. Why was this interval required? The answer to this question might shed light on why the 3 day interval was more successful at generating articular cartilage than was the 7 day interval.

4. Articular cartilage formation in the BMP9 group was observed in 51% of the treated animals. Why did 49% of them fail to form articular cartilage? When the sample was considered a failure, what tissue did form? What is the articular cartilage formation rate in BSA-treated control group?

5. It would be helpful to show a side-by-side histologic comparison of BMP9-treated and control (untreated) digit tips at PN35. MicroCT imaging shows a gap shown in the BMP9-treated digits; without histology it is not possible to determine if the gap is filled by fibrous tissue or by a regenerated joint space.

Minor points:

1. Fig 1A is too small to show the positive cells.
2. The labels in Fig 2H and I are not correct.
2. The title for Fig 2 should be modified.
4. Fig 3D needs a label.
5. Fig 3B-M could use some clarification: are these samples from the 3-day interval group or 7-day interval group?

Response to reviewer comments:

Reviewer #1

This MS contends that BMP9 treatment of digit amputation wounds in mice stimulates regeneration of synovial joint structures that articulate with the stump bone. As previous studies showed that treatment by other BMPs induced endochondral bone formation but did not result in regeneration of joint structures, this finding, if accurate, represents a novel and perhaps clinically significant finding. However, while interesting, the data provided are incomplete and many of the appropriate controls required to verify the specific actions of BMP9 are missing. In addition, the authors provide no mechanism by which BMP9, unlike other BMPs, would be able to initiate synovial joint regeneration.

Specific comments:

1. The ISH for Bmp9 are really difficult to see; Fig 1A shows many areas of potential Bmp9 expression throughout the section and nothing that appears specific to the interzone. As Bmp9 is not expressed in these other areas, it is possible that the reported interzone staining is an artifact. As Bmp9 KO mice have no skeletal phenotype, Bmp9 is certainly not required for digit joint formation. The authors are encouraged to use other validation methods (laser capture micro-dissection followed by qPCR for example) to verify the localization of Bmp9 in the digit joint.

The counterstain we use can obscure the in situ signal when reproduced for publication. We have replaced the image in Figure 1A with one that has not been counter stained. The in situ expression of Bmp9 is specific to joint interzone cells that are undergoing a cavitation response. This expression is not an artifact. We have also replaced supplemental figure 1C at E15.5 with an in situ image that lacks counter staining and interzone specific expression of Bmp9 is also demonstrated at this earlier stage. All of the images have been enlarged for clarity.

2. The authors state that they see joint regeneration 51% of the time suggesting the repair observed is quite variable. The authors should show the best and worst case scenarios and provide the criteria they used to score each of the digit regenerates. A complete time course comparing control to BMP9 treated digit stumps should also be included so that the full picture of BMP9 activity can be evaluated. As BMP9 is known as a potent bone formation agent, more potent than BMP2, the authors should comment on whether there was any ectopic bone formation that occurred.

The joint regeneration response is observed in about half of the treated samples. We use MicroCT to initially evaluate the digits and they are scored based on the regeneration of a distinct distal bone. Subsequent histological evaluation shows that the regenerated bones are not ectopic but form an articulation with the stump bone, so we have used the term “nodule” to distinguish them from ectopic bones induced in soft tissue by BMP application. We have added a supplemental figure (SFig. 2) to show a complete time course of the response. The BMP9 response is divided into cavity forming digits (~50%) and non-cavity forming digits (~50%) and these are compared to BSA controls. SFig. 2A shows an example of a worst case scenario for an identified joint structure as suggested. We have added a brief explanation of how digits were scored (lines 61-65).

3. For the sequential bead implantation studies, several additional controls should be included; BMP2 followed by BMP2; BMP9 followed by BMP9; and those should be compared to BMP2 followed by BMP9 and BMP9 followed by BMP2. These data would solidify the unique role of BMP9 and remove the possibility that continuous BMP activity, unrelated to BMP9, is what drives the regenerate.

We have carried out the suggested studies and some of the data have been added to the manuscript. The sequential implantation of BMP2 followed by BMP2 is an important control and these digits show an enhancement of stump bone regeneration but no regeneration of joint structures. We have added these data to the manuscript (Fig. 4D and lines 193-197). We have also carried out the control of BMP9 treatment later in the healing process, i.e. withholding the initial BMP2 treatments, and these digits do not regenerate articular cartilage on the stump bone (Fig. 4E,F). We have previously shown that BMP2 treatment at this late stage in the healing process fails to elicit a regenerative response (Dawson et al., 2017) and the data suggests that BMP2 and BMP9 are affecting distinct cell populations. The suggested studies of BMP9 followed by BMP9 and BMP9 followed by BMP2 are not appropriate controls for this study so we have not included the results. We have also added a paragraph that consolidates evidence showing that the BMP9 response is uniquely distinct from BMP2 in this digit amputation model (lines 98-111).

4. The authors provide no mechanism for why BMP9 might have a unique activity but do suggest it might be due to utilization of Alk1 and Acvr2B. Since there are good reagents to localize these receptors, and there are cKO mice as well, the authors could explore their hypothesis by localizing the receptors during the regeneration process and demonstrating regeneration doesn't occur in their absence or when other BMPs are implanted.

We have revised the manuscript to address the mechanism of BMP9 action. The original manuscript described BMP9 regeneration of joints and we did not emphasize a mechanism of action. In the revised manuscript we emphasize data showing that BMP9 induces two independent responses, chondrogenesis and cavitation, both of which are required for the regeneration of joint structures. This is observed in comparing the cavity forming digits (Fig. 1F-L) to the non-cavity forming digits (Fig. 1R-V) following BMP9 treatment (lines 82-98). We have also added new data using Prg4 mutant digits that show chondrogenic gene expression is induced by BMP9 in the absence of Prg4 (Fig. 2I-M, lines 144-153). These studies provide functional evidence that there are two independent BMP9 responses and that Prg4 is required for only one: the cavitation response. We do not propose that the BMP9 response utilizes Alk1 and Acvr2B, only that studies on other cell types show that BMP9 is a ligand for this receptor complex. Our attempts to localize Alk1 to the amputation wound with immunohistochemistry, in situ hybridization, microarray or qPCR have been negative. While cKO mice are available and the question of receptor function is relevant, we will not be in a position to design an experimental approach until we uncover the cell type or types that respond to BMP9. The mechanism we propose is cell-based and consistent with recent studies showing that Prg4 expressing cells of the developing synovial cavity becomes the superficial layer of articular cartilage and contains stem cells that invade underlying chondrogenic tissue to form all articular cartilage layers (Kozhemyakina et al 2016; Li et al 2017). This is proposed on lines 241-248 of the revised manuscript.

Reviewer #2 (Remarks to the Author):

This paper continues with this group's previous studies on how to stimulate regeneration of digits (and lower limbs) in mice. They have previously shown that a P2 amputation of a digit does not regenerate anything but if BMP2 is added on a bead then growth of the phalange is stimulated and new endochondral bone is formed. There is no new joint, however. This work shows that by the subsequent addition of BMP9 (after BMP2) a new joint can be produced in addition to the elongation of P2. Making a new joint is a major achievement and represents a highly significant advance by this group. The joint is demonstrably so because of i) particular cartilage on both the end of P2 and the 'nodule'; ii) the expression of several joint markers (Fmod, Ucn3, Acan, Dcx); iii) the failure to form in a Prg4 mutant mouse. I think this work should be published in Nature Comm. I have a few comments on the data.

1. Have these BMP2/BMP9 experiments been done on adults which would be directly relevant to adult human digit amputations? In this work the digits are amputated on PN3.

We have carried out BMP9 studies in adult P2 digit amputations using a sol-gel delivery protocol that is effective for BMP2 induced skeletal regeneration (Dawson et al., 2017) and the data has been added to the manuscript (Fig 3, lines 154-177). Treatment of adult amputations with BMP9 induces major chondrogenic outgrowth and we observe a cavitation response at a low frequency (20%) 14 days after treatment. The data suggest that the adult amputation wound is biased for the chondrogenic BMP9 response but that the cavitation response can occur.

2. line 54/55. In Fig.1F there is articular cartilage on the nodule. Why is there no articular cartilage on the nodule in Fig. 1G ? Is it because of the word 'occasionally' in line 53?

The digit in 1F is 9 weeks after BMP9 treatment and the articular cartilage is mature and histologically identifiable. The digit in 1G is 14 days after BMP9 treatment and is immature and at that stage articular chondrocytes cannot be identified by histology. The word occasionally in line 53 refers to the stump response in mature digits where we occasionally observe patches of articular chondrocytes and not a layer of articular cartilage.

3. Most of the in situs show very clear signals, but a few of them do not. I find it very difficult to see the signal in SFig 1N & O (Fmod & Ucma respectively).

In the revised manuscript we have enlarged the images and reduced the background staining to enhance the signal. SFig 1N & O of the original manuscript are Fig 1P & Q of the revised manuscript. These are control BSA treated digits with no signal in the amputation wound and arrowheads point to a low level of expression in the P2 joint.

4. lines 83/84. Fmod and Ucma seem to be up-regulated throughout the nodule rather than just in the cells on the joint side of the nodule. Is the whole nodule a joint or is this what happens in the development of a joint?

The entire nodule is not a joint. It is a chondrogenic aggregate that is necessary for joint regeneration but a cavitation response is also required. The relationship between Fmod and Ucma to articular chondrocytes is not clear. Based on the available literature we identified a number of genes that are expressed in cartilage of the joint region where articular cartilage forms and used these genes as markers of articular chondrocytes. Cell lineage studies indicate that superficial cells invade this cartilage to form all layers of articular cartilage (Kozhemyakina et al 2016; Li et al 2017) so it is likely that the articular cartilage marker genes that we used more accurately identify cartilage that is later invaded by articular cartilage stem cells. In the revised manuscript we have been careful to identify this tissue as cartilage and not articular cartilage.

5. Fig. 1P - there looks to be 1 Dcx +ve cell on the P2 side of the joint, agreed it is not on the endochondral surface. Are there non joint cells that express Dcx?

The expression of Dcx during joint development is restricted to chondrogenic tissues where articular cartilage forms. For the most part, we observe Dcx expression associated with superficial cells of the BMP9 induced joint, however we do observe some Dcx positive cells that are internal. The single cell on

the P2 side of the joint is one example and we also observe some internal cells within the regenerated nodule. We note this in the figure legend (line 336-337)

6. As you read this you get the impression that BMP2 is going to lengthen the P2 phalange, then BMP9 is going to make a joint. So the final product should be quite a bit longer than either a control (nothing) or a BMP9 only (just a joint) regenerate. But then when I compare Fig. 1C with Fig. 3B they both look the same. Have I got the wrong impression or is there not an overall difference in these varying regenerates?

You do not have the wrong impression but there is variability in digit length between samples. We have measured the stump length of B2-B9 treated digits and they are significantly longer when compared to controls. We have added these data to the revised manuscript (line 209-211).

Reviewer #3 (Remarks to the Author):

Previous work by this group has demonstrated that digit tip regeneration can be induced in the early post-natal mouse with individual morphogenetic agents (e.g., BMP2 and BMP7), which initiate a “multi-tissue response that culminates in structural regeneration” (Stem Cells Transl. Med., 2018). The goal in this manuscript was to test whether BMPs could induce both bone and cartilage formation and in doing so, regenerate the joint space. The basis of this work is, I believe, predicated on Urist’s findings in the early 1970’s that BMPs induce chondrogenesis. Here, the authors use a BMP9 soaked bead to also induce chondrogenesis, this time at the digit tip. I don’t think its correct to state, “This (chondrogenic) response is specific to BMP9 treatment” since they (and others) have shown that other BMPs can also induce chondrogenesis (e.g., Yu et al., Dev Biol, 2012). What is new, however, is the observation that BMP2 and BMP9 have differing effects on the wound blastema: where BMP2 induces the formation of cartilage that undergoes hypertrophy, BMP9 induced cartilage does not.

Thank you for pointing out our mis-statement. We were trying to convey that the combined formation of a chondrogenic response and a cavitation response is unique to BMP9. In the revised manuscript we have been careful to note differences between BMP2 and BMP9 without making sweeping statements.

What is missing from this otherwise compelling study is a clear mechanism of action to explain the observed phenomenon. Other investigators have shown that BMP9 induces hypertrophic-like state with increased collagen type X expression in chondrocytes (van Caam et al. 2015), and that BMP9 can directly induce osteogenesis (Fujioka-Kobayashi et al. 2018). Presumably the cellular response to BMP9 depends in part on the identit(ies) of the cells on which it is acting. Which cells are responding to BMP9 in the digit anlagen? Given that chondroblasts and osteoblasts arise from a common progenitor (e.g., see Nature Cell Biology (Ono et al. 2014), it is unclear how the authors’ findings align with the current literature. The experiments conducted with the Prg4 mutant strain were not particularly informative with regards to a mechanism of action. There are other questions (outlined below) that if answered, would significantly raise enthusiasm for this interesting paper.

In the revised manuscript we point out that our findings do align with current literature on how articular cartilage develops and in this regard we believe that the Prg4 mutant studies are informative with respect to mechanism. There are many types of chondrocytes that have evolved to serve distinct functions and studies show that articular chondrocytes do not represent a default state, but form from precursor cells present within the Prg4 expressing layer that line the synovial cavity that invade the underlying cartilage to form all layers of articular cartilage (Kozhemyakina et al 2016; Li et al 2017). These findings are consistent with Col2 expressing lineage studies by Ono et al. (2014) that suggest adult articular cartilage is

derived from superficial cells of the neonate. We propose a similar mechanism for joint regeneration that involves chondrogenic induction in association with cavitation to establish a superficial layer of Prg4 expressing cells that invade the underlying cartilage to regenerate articular cartilage (line 241-248).

1. The authors should provide some justification as to why BMP9 was chosen to stimulate joint formation.

We have modified the introductory paragraph to better explain why BMP9 was studied for joint regeneration (line 41-47). Briefly, Bmp9 is expressed during joint cavitation, joint formation is inhibited in Noggin mutants and BMP9 lacks a Noggin binding domain.

2. Given the clinical side effects of BMP use (Carragee 2014), it would be helpful to include information on dose-dependent effects in this developmental context.

We have now addressed the dose of BMP9 administered per digit. This is presented in SFig. 1F and is estimated at 5.5 ng delivered over 72 hours. We used a BMP responsive cell line BRITER in the presence of Noggin to generate a dose-response curve and estimated timed release in vitro. This delivered dose is sufficiently low so as to not cause significant side effects and we have not observed evidence of any side effects.

3. Questions about dose: why did the authors choose a bead soaking concentration of 0.5ug/ul? What was the actual dose delivered to the tissues? What is the half-life of BMP2? The reason why this latter question is important to answer is based on the observation that sequential treatment with BMP2 followed by BMP9 appeared to require an interval of 3 or 7 days. Why was this interval required? The answer to this question might shed light on why the 3 day interval was more successful at generating articular cartilage than was the 7 day interval.

Agarose bead delivery of purified growth factors has been used to investigate transient effects in many developmental and regenerative studies. Soaking concentrations usually range for 1.0 to 0.5 ug/ul and these values are established empirically. We do not have data on in vivo dose-dependent effects and it would be impossible to gather such data based on induced anatomical changes such as joint regeneration. In a previous study (Yu et al., 2014, Angiogenesis is inhibitory for mammalian digit regeneration) we have shown dose-dependent inhibition of regeneration using bead delivery of BMP9 (soaking concentrations of 0.5, 0.25, 0.05 and 0.01 ug/ul), so this delivery method does display a dose-response. As mentioned, the actual dose delivered in vivo is estimated at 5.5 ng delivered over a 72 hour period. We do not have information on the half-life of BMP2, however we have used a constant dose of BMP2 delivered at different times during the healing process to characterize the response window (Dawson et al., 2017), and we show that the regeneration response is lost within 3 days of application. This is consistent with our measured release data (SFig. 1F) and suggest that the effective local half-life of BMP2 is 2-3 days. The 3 or 7 day interval for our B2-B9 studies was selected because that is the period following BMP2 treatment when an endochondral ossification center can be identified (Yu et al., 2012; lines 181-185). The reviewer is correct that this time frame provides hints about the response and our findings show that the longer interval of 7 days was more successful than the 3 day interval. These results provide clues about the potential cell types that are responsive to BMP9 but we have not yet identified those cell type(s).

4. Articular cartilage formation in the BMP9 group was observed in 51% of the treated animals. Why did 49% of them fail to form articular cartilage? When the sample was considered a failure, what tissue did form? What is the articular cartilage formation rate in BSA-treated control group?

In the revised manuscript we have studied the non-cavity forming digits (Fig. 1R-V) following BMP9 treatment (lines 90-95) and show that a chondrogenic response is stimulated by BMP9. We have also added new data using Prg4 mutant digits that show that chondrogenic gene expression is induced by BMP9 without cavity formation (Fig. 2I-M, lines 144-152). These studies provide functional evidence that there are two independent BMP9 responses necessary for stimulating joint and articular cartilage regeneration, and that Prg4 is required for only one: the cavitation response. In the 49% that fail, we observe a small aggregation of chondrocytes at the end of the stump that appears to regress, so the digits appear similar to control digits by 14 days (Sfig. 2). Control BSA-treated digits never form articular cartilage.

5. It would be helpful to show a side-by-side histologic comparison of BMP9-treated and control (untreated) digit tips at PN35. MicroCT imaging shows a gap shown in the BMP9- treated digits; without histology it is not possible to determine if the gap is filled by fibrous tissue or by a regenerated joint space.

We have added a figure (Sfig. 2) that shows side-by-side histology of BMP9 and control digits at different stages. We have never observed fibrous tissue filling the gap between the stump and the regenerated skeletal element.

Minor points:

1. Fig 1A is too small to show the positive cells.

We have enlarged all of the images for clarity.

2. The labels in Fig 2H and I are not correct.

Labels have been corrected.

2. The title for Fig 2 should be modified.

We have changed the title to “Prg4 is required for BMP9-induced cavitation”

4. Fig 3D needs a label.

This has been corrected

5. Fig 3B-M could use some clarification: are these samples from the 3-day interval group or 7-day interval group?

All of the samples are from the 3 day interval study and we have added that information to the figure legend (now Figure 4).

Reviewers' Comments:

Reviewer #1:

Remarks to the Author:

The authors present an intriguing finding about a role for BMP9 in regeneration. Although the authors added more supplemental data and enhanced several of the photographs, several issues remain to be resolved:

1. There is no explanation for why the repair phenotype is present only in 50% of the digits.
2. As joints lacking BMP9 undergo cavitation, BMP9 cannot be required for this process. In adults, BMP9 is present in circulation so would be available to the blastema during amputation healing, and there would be greater amounts of BMP9 during wound repair due to the increased vascularity. As BMP9 infrequently induces cavitation in the authors' model, it is difficult to conclude that BMP9-mediated cavitation is a fundamental step in regeneration.
3. BMP9 induces endochondral ossification and enhances fracture repair; these processes both require chondrocyte hypertrophy and others have posited that the lack of noggin antagonism of BMP9 makes it a more potent BMP when compared to BMP2 and more potent inducer of hypertrophy. In the authors' regeneration model, BMP9 does not induce hypertrophy while BMP2 does. No real explanation is given for this curious finding and the authors state that mechanistically, this is a paper about cells and not signaling, yet this difference seems to be about signaling as the cell population is the same.

Reviewer #2:

Remarks to the Author:

The authors have responded to my comments by adding text, altering figures where relevant and adding a good deal of new data on the response of adults to BMPs (Fig. 3).

I think this is now suitable for publication.

Reviewer #3:

Remarks to the Author:

My questions have all been addressed, and the answers were both informative and thorough.

The authors present an intriguing finding about a role for BMP9 in regeneration. Although the authors added more supplemental data and enhanced several of the photographs, several issues remain to be resolved:

1. There is no explanation for why the repair phenotype is present only in 50% of the digits.

The mammalian joint is a complex structure that is non-regenerative after injury. BMP9 induces joint regeneration with a frequency of 50-70% and we have added a statistical assessment to the revised manuscript (Line 63-64) to emphasize that induced joint regeneration is significant. BMP9 initiates two distinct responses during joint regeneration: a chondrogenic response observed in 100% of the digits and a cavitation response observed in approximately half of the digits. In the revised manuscript we have compiled histological data independent of MicroCT imaging to determine the frequency of cavity formation, and this was found to be 61% (60/98). These data have been added to the revised manuscript (Line 63-67). The similarity between the frequency of joint regeneration and induced cavitation is suggestive of a causal link. Cells that form the cavity express Prg4, and Prg4 mutant studies indicate that both joint regeneration and cavitation induced by BMP9 requires Prg4. Alternatively, BMP9 induces chondrogenesis independent of Prg4 indicating two different BMP9 responsive cell populations, and identifies the Prg4 expressing cell population as rate-limiting for joint regeneration. We also show that the frequency of joint regeneration is significantly increased to 70% in B2-B9 treated digits (Line 193-194), showing that BMP2 treatment enhances the BMP9 response. We have added new data to the revised manuscript (Line 207-215; SFig. 3O) showing that BMP2 treatment induces Prg4 expression but without inducing cavity formation. This result indicates that Prg4 expression is required but not sufficient to induce cavitation and joint regeneration, and suggests that BMP2 enhances the BMP9 response by enriching the Prg4 expressing cell population. All of the evidence points to BMP9-induced cavitation as the rate-limiting step in the response and this is “why the repair phenotype is present only in 50% of the digits.”

2. As joints lacking BMP9 undergo cavitation, BMP9 cannot be required for this process. In adults, BMP9 is present in circulation so would be available to the blastema during amputation healing, and there would be greater amounts of BMP9 during wound repair due to the increased vascularity. As BMP9 infrequently induces cavitation in the authors' model, it is difficult to conclude that BMP9-mediated cavitation is a fundamental step in regeneration.

*Mammals including humans display very limited regenerative capabilities and there are currently no models for joint regeneration, endogenous or induced, that can be explored to identify fundamental steps. BMP9-induced joint regeneration represents the first mammalian model of joint regeneration, and we understand that the reader might draw the conclusion that “BMP9-mediated cavitation is a fundamental step in regeneration”, but it is critically important to point out that we do not make this conclusion. While it is difficult to imagine synovial joint regeneration without forming a synovial cavity, it is a certainty that synovial cavity formation can occur in the absence of Bmp9. However, it is also important to note that redundancy of gene function is common in joint and skeletal development, and that induced regeneration has been demonstrated with factors that are not individually required for development. We have modified the revised manuscript to make note of this (Line 237-240). Lastly, the speculative scenario of circulating BMP9 influencing blastema formation in wound repair is interesting, however one of the characteristics of the regeneration blastema is that it is avascular and hypoxic, making it a site that can develop in isolation of systemic influences (Fernando et al., 2011, *Develop. Biol.* 350, 301 PMID: 21145316; Sammarco et al., 2014, *J Bone Miner Res.* 29(11): 2336 PMID: 24753124.; Sammarco et al., 2015, *PLoS ONE* 10(10): e0140156. doi:10.1371/journal.pone.0140156 PMID: 26452224; Yu et al., 2014, *Regeneration* 1(3), 33 PMID: 27499862). So, the accumulation of circulating BMP9 in the blastema is not predicted.*

3. BMP9 induces endochondral ossification and enhances fracture repair; these processes both require chondrocyte hypertrophy and others have posited that the lack of noggin antagonism of BMP9 makes it a more potent BMP when compared to BMP2 and more potent inducer of hypertrophy. In the authors' regeneration model, BMP9 does not induce hypertrophy while BMP2 does. No real explanation is given

for this curious finding and the authors state that mechanistically, this is a paper about cells and not signaling, yet this difference seems to be about signaling as the cell population is the same.

*In some models, for example induced osteogenesis, BMP2 and BMP9 stimulate a similar response that can be quantified and the lack of noggin antagonism helps to explain why BMP9 is more potent than BMP2. In other models BMP9 and BMP2 responses are not similar, for example induced angiogenesis, where the two factors effect cells in distinct ways. There are aspects of the digit amputation model where BMP2 and BMP9 stimulate a similar response, e.g. chondrogenesis, but other aspects where the two responses are dissimilar, e.g. cavity formation. Since the cavitation response tracts with joint regeneration we have focused attention on BMP9-induced cavitation and less so on BMP9-induced chondrogenesis. We present data on chondrocyte hypertrophy in the context of demonstrating that BMP9-induced joint regeneration and BMP2-induced skeletal regeneration can be stimulated together. The statement “**BMP9 does not induce hypertrophy while BMP2 does.**” is not accurate. As we show BMP9 stimulates hypertrophic chondrocyte differentiation 14 days after treatment (SFig. 2J) and we have now added new data showing ColX expressing cells 11 days after treatment (SFig. 3G). In previous studies, BMP2 induces Col2a1 expressing chondrocytes within the 3 day treatment period and Col10a1 expression is delayed until 2 days after the BMP2 bead is exhausted (Yu et al., 2012). The data indicate that BMP2 induces initiation of a chondrogenic program that results in hypertrophy, but does not induce hypertrophy. Our data with BMP9 is similar but varies in the length of time to the onset of hypertrophy. In experiments testing the effect of BMP2-BMP9 sequential treatment, BMP2 stimulated hypertrophy of digit stump cells is delayed from 5 days to 7 days after BMP2 treatment, and this delay coincides with the period of BMP9 treatment. This suggest that BMP9 has an inhibitory effect on hypertrophy and this is consistent with the time delay associated with BMP9 treatment alone. We have modified the text in the revised manuscript to clarify the effect that BMP9 has on hypertrophic chondrocyte differentiation (Line 103-107), differences between BMP2 and BMP9 treatments (Line 112-114), and the effect of BMP9 on Col10a1 expression in BMP2 treated digits (Line 215-222).*

*The critique “**the authors state that mechanistically, this is a paper about cells and not signaling**” is not accurate: we don’t make such a statement. The data lead us to the conclusion that some cells respond to BMP9 treatment by undergoing chondrogenic condensation while other cells respond by undergoing cavitation. Both responses are required for BMP9-induced joint regeneration: the chondrogenic response is required to regenerate the distal skeletal element and the cavitation response is required to regenerate the synovial cavity. Clearly BMP9 signaling plays a key role, but the evidence is not consistent with a simple signaling model in which BMP9 signaling stimulates a single cell type that regenerates a joint. Instead, BMP9 is acting directly or indirectly on at least two different cell types to effect joint regeneration and there are many potential mechanistic scenarios all involving BMP9 signaling. The importance of this manuscript is demonstrating for the first time that mammalian joint regeneration can be stimulated and that BMP9 plays a role in this response. This is a complex regenerative response and identification of essential signaling pathways will require identification of the responsive progenitor cells.*

Reviewers' Comments:

Reviewer #1:

Remarks to the Author:

The authors have addressed all of my concerns but one, that BMP9 is actually expressed in the developing joint. As that fact does not really enter into the blastema and regeneration story they present, and they have not done the suggested controls to prove the BMP9 localization, I suggest they remove that panel.

Response to Referee 1:

The authors have addressed all of my concerns but one, that BMP9 is actually expressed in the developing joint. As that fact does not really enter into the blastema and regeneration story they present, and they have not done the suggested controls to prove the BMP9 localization, I suggest they remove that panel.

*Referee 1 suggest removal of the panel (Fig 1a) showing that Bmp9 is expressed in the developing joint, however this will greatly diminish the quality of the paper and is contrary to Referee 3's suggestion to provide a **“justification as to why BMP9 was chosen to stimulate joint formation”**.*

*In our original submission, Referee 1 noted that **“ISH for Bmp9 are really difficult to see”** and suggested that we use laser microdissection and qPCR as an alternative method. The in situ hybridization signal in images reproduced for publication can be obscured by the counterstaining procedure we use to visualize surrounding tissues and Referee 1 was mistaking the in situ signal with the counterstain. In response, we repeated our in situ hybridization studies but omitted the counterstaining procedure and this resulted in a clear in situ signal identifying Bmp9 transcripts in cells forming the synovial cavity (Fig 1A). As positive controls we show that Bmp9 transcripts localize to the developing liver known to express Bmp9 (based on RT-PCR) (Sup Fig 1A), and that Bmp9 is expressed at multiple stages of digit joint development (Sup Fig 1C). As negative controls we show that Bmp9 is not expressed in the limb prior to joint formation (Sup Fig 1B), and Bmp9 transcripts are absent after joint formation is complete (Sup Fig 1d). Our in situ hybridization studies clearly demonstrate that Bmp9 is specifically expressed during joint development and this is supported by comments by Referees 2.*

Referee 1's suggestion to use laser microdissection and qPCR is, in fact, not a control for in situ hybridization as suggested, but an alternative method if in situ hybridization was technically impossible. However, laser microdissection and qPCR lacks the precision required to prove that transcripts localize to cells involved in morphogenesis.

*Referee 1 also states that the expression of Bmp9 during joint development **“does not really enter into the blastema and regeneration story”**. We disagree strongly with this statement because the expression of Bmp9 during joint morphogenesis establishes the rationale for testing its role in regeneration. Indeed, this aspect of the manuscript was developed in the 1st revision in response to comments from Referee 3 who asked for a justification for testing BMP9 in regeneration.*

For these reasons, we believe that following Referee 1's suggestion to remove Fig 1A would weaken the manuscript, and we have retained this panel in the revised manuscript.